# SLAMF6 deficiency augments tumor killing and skews toward an effector phenotype revealing it as a novel T cell checkpoint

Emma Hajaj[1,2,3]*, Galit Eisenberg[1,2], Shiri Klein[1,2], Shoshana Frankenburg[1,2], Sharon Merims[1,2], Inna Ben David[1,2], Thomas Eisenhaure[4], Sarah E Henrickson[4,5†‡], Alexandra Chloé Villani[4,6,7,8], Nir Hacohen[4,6,7], Nathalie Abudi[2,9], Rinat Abramovich[2,9], Jonathan E Cohen[1,2], Tamar Peretz[1], Andre Veillette[10], Michal Lotem[1,2,3]

[1]Sharett Institute of Oncology, Hadassah Hebrew University Hospital, Jerusalem, Israel; [2]Wohl Institute for Translational Medicine, Hadassah Medical Organization, Jerusalem, Israel; [3]Lautenberg Center for Immunology and Cancer Research, Faculty of Medicine, Hebrew University, Jerusalem, Israel; [4]Broad Institute of MIT and Harvard, Cambridge, United States; [5]Boston Children's Hospital, Department of Pediatrics, Boston, United States; [6]Center for Cancer Research, Massachusetts General Hospital, Charlestown, United States; [7]Department of Medicine, Harvard Medical School, Boston, United States; [8]Center for Immunology and Inflammatory Diseases, Massachusetts General Hospital, Charlestown, United States; [9]Goldyne Savad Institute of Gene Therapy, Hadassah Hebrew University Hospital, Jerusalem, Israel; [10]IRCM, Montreal Clinical Research Institute, Montreal, Canada

*For correspondence:
emma.hajaj@mail.huji.ac.il

Present address: †Institute for Immunology, Perelman School of Medicine, University of Pennsylvania, Pennsylvania, United States; ‡Department of Pediatrics, Division of Allergy and Immunology, The Children's Hospital of Philadelphia, Philadelphia, United States

Competing interests: The authors declare that no competing interests exist.

**Abstract** SLAMF6 is a homotypic receptor of the Ig-superfamily whose exact role in immune modulation has remained elusive. Its constitutive expression on resting and activated T cells precludes it from being a *bona fide* exhaustion marker. By breeding Pmel-1 mice with SLAMF6 -/- mice, we generated donors for T cells lacking SLAMF6 and expressing a transgenic TCR for gp100-melanoma antigen. Activated Pmel-1xSLAMF6 -/- CD8+ T cells displayed improved polyfunctionality and strong tumor cytolysis. T-bet was the dominant transcription factor in Pmel-1 x SLAMF6 -/- cells, and upon activation, they acquired an effector-memory phenotype. Adoptive transfer of Pmel-1 x SLAMF6 -/- T cells to melanoma-bearing mice resulted in lasting tumor regression in contrast to temporary responses achieved with Pmel-1 T cells. LAG-3 expression was elevated in the SLAMF6 -/- cells, and the addition of the LAG-3-blocking antibody to the adoptive transfer protocol improved the SLAMF6 -/- T cells and expedited the antitumor response even further. The results from this study support the notion that SLAMF6 is an inhibitory immune receptor whose absence enables powerful CD8+ T cells to eradicate tumors.

## Introduction

The SLAM family of receptors (SFRs) is a set of six receptors expressed on hematopoietic cells (*Wu and Veillette, 2016*; *Cannons et al., 2011*; *Veillette, 2010*; *Calpe et al., 2008*). All SFRs, except 2B4, are homotypic binders, that is they engage the same ectodomain sequence, either in cis (same cell) or in trans (adjacent cell) configuration. Most hematopoietic cell types express 3–5 members of the SLAM family.

**eLife digest** The immune system helps to protect our bodies from illnesses and infections. Immunotherapies are medicines designed to treat diseases, such as cancer, by boosting the immune system against the condition. This is a powerful approach but so far immunotherapies have only had partial success and there is a need for further improvements.

One protein called SLAMF6 is found on cells from the immune system that attack and kill cancer cells. Immunotherapies that suppress SLAMF6 on immune cells called killer T cells could increase immune system activity helping to treat cancers, particularly melanoma skin cancers. So far the potential for SLAMF6 as a target for immunotherapy has not been fully explored.

Hajaj et al. created mice with killer T cells that recognized skin cancer cells and lacked SLAMF6. These modified cells were better at fighting cancer, producing more anti-cancer chemicals called cytokines and killing more cancer cells. The modified cells had a lasting effect on tumors and helped the mice to live longer. The effects could be further boosted by treating the mice in combination with other immunotherapies.

SLAMF6 is a possible new target for skin cancer immunotherapy that could help more people to live longer following cancer diagnosis. The next step is to create a drug to target SLAMF6 in humans and to test it in clinical trials.

SFRs generate signals via a bi-phasic mechanism of recruitment to tyrosines in the immunoreceptor tyrosine-based switch motifs (ITSMs) in their cytoplasmic domain. SLAM associated protein (SAP), a small protein containing the Src homology 2 (SH2)-domain, was shown to be the default adaptor of the SFRs, interchanging with protein tyrosine phosphatases, mainly SHP-1, but also SHP-2, inositol phosphatase SHIP-1 and protein tyrosine kinase Csk (*Wu and Veillette, 2016*; *Cannons et al., 2011*; *Veillette, 2010*; *Calpe et al., 2008*).

SLAMF6, also known as CD352, LY108, or NTB-A, is a homotypic SFR expressed on T cells, NK cells, B cells, and dendritic cells (*Bottino et al., 2001*; *Zhong and Veillette, 2008*). Kageyama et al. linked SLAMF6 to the anchoring of T cells to their target cells, and subsequent cytolysis of the target (*Kageyama et al., 2012*). According to these authors, functional SAP is critical for SLAMF6 activity. In mice lacking SAP, SLAMF6 was shown to inhibit T cell function (*Kageyama et al., 2012*; *Zhao et al., 2012*; *Bottino et al., 2001*). The role of SLAMF6 in healthy T cells expressing normal SAP levels was generally inferred from contextual data and is not yet clear. There are indications that SLAMF6 plays an activating role in double-positive thymocytes (*Dutta et al., 2013*) along with evidence that it plays an inhibitory role in iNKT cells and CD8+ T cells (*Lu et al., 2019*; *Eisenberg et al., 2018*). Gene expression profiles of T cell subsets link SLAMF6 to the progenitor-exhausted state (*Miller et al., 2019*) and to the tuning of the critical number of T cells required for proper differentiation (*Polonsky et al., 2018*).

To elucidate the net function of SLAMF6, we generated a transgenic mouse with the Pmel-1 melanoma-specific T-cell receptor (TCR) expressed in CD8+ T cells, in which the *SLAMF6* gene was knocked out. In this report, we show for the first time that SLAMF6 -/- CD8+ T cells display improved anti-melanoma activity and prevent melanoma growth more effectively than CD8+ T cells with intact and functional SLAMF6. Since SLAMF6 is constitutively expressed on T cells, it acts as an inhibitory checkpoint receptor whose absence allows the eradication of established tumors by CD8+ T cells.

## Results

### SLAMF6 is constitutively expressed on T cells and increases upon activation

SLAMF6 is an immune receptor constitutively expressed on non-activated and activated T cells (*Eisenberg et al., 2018*). The level of SLAMF6 transcription and receptor expression, however, is dynamic, changing with time and activation states. To record SLAMF6 expression in a longitudinal manner, human tumor-infiltrating lymphocytes (TILs) were activated for 5 days, and SLAMF6 transcript and protein expression were measured (*Figure 1A–C*). After 1 day of activation, there was an initial decrease in the SLAMF6 transcript that switched to over-expression (*Figure 1C*). From 3 days

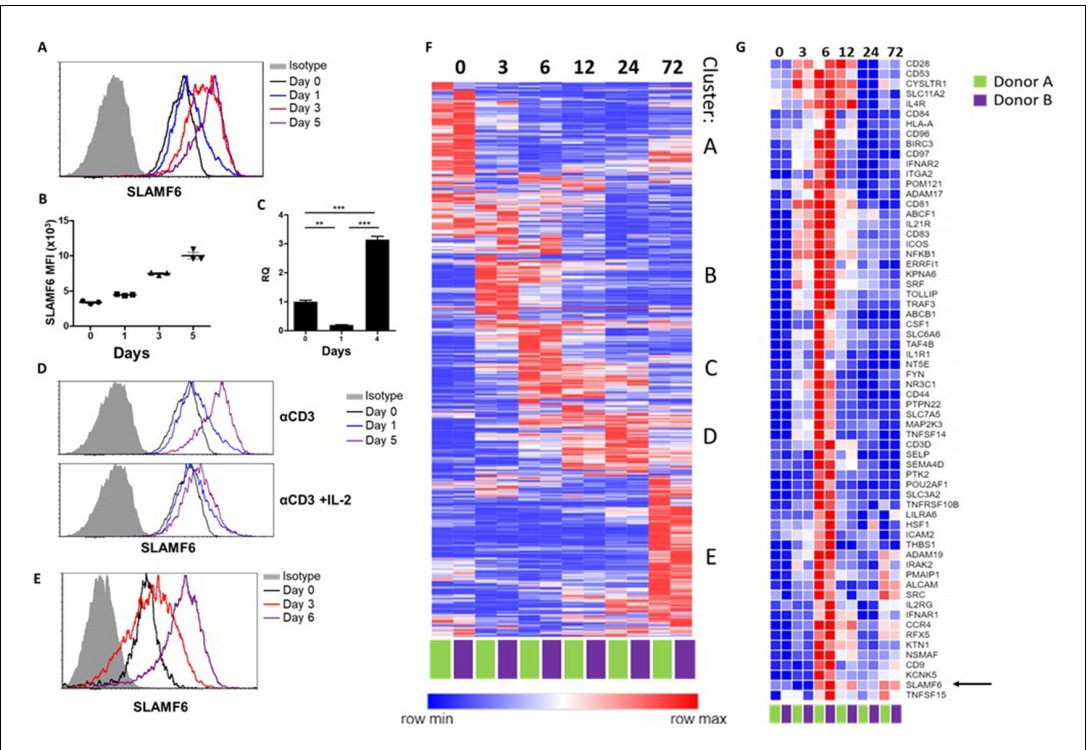

**Figure 1.** SLAMF6 is constitutively expressed on T cells and increases upon activation. (**A–C**) SLAMF6 expression in human TIL412 cells, activated for five days. (**A**) Flow cytometry at the indicated time points. (**B**) Median fluorescence intensity (MFI) of SLAMF6, days 1–5. (**C**) Quantitative RT-PCR for *SLAMF6*. RNA was extracted at the indicated time points. Data normalized to *HPRT* expression at each time point and to the basal expression level on day 0. One-way ANOVA. **, p<0.01, ***, p<0.001. (**D**) SLAMF6 expression by flow cytometry in human TIL412 cells activated for 5 days with anti-CD3 or with anti-CD3 plus IL-2, at the indicated time points. (**E**) SLAMF6 expression by flow cytometry in Pmel-1 mouse splenocytes activated for 6 days, at the indicated time points. (**F**) Row normalized expression of immune-related genes from RNAseq, clustered according to similar expression patterns. CD4+ T cells from two donors were stimulated with anti-CD3 plus anti-CD28 for 72 hr, RNA was extracted and sequenced. Numbers in the top panel indicate hours. (**G**) Magnification of cluster C. *SLAMF6* is marked.

The online version of this article includes the following source data for figure 1:

**Source data 1.** RNA sequencing of healthy donors CD4 T cells along activation.

after activation onward, SLAMF6 receptor expression consistently increased (*Figure 1A and B*). Interestingly, the increased expression was most pronounced in T cells activated in the absence of IL-2 (*Figure 1D*). A similar pattern was observed for the expression of the murine SLAMF6 receptor on Pmel-1 CD8+ T cells (*Figure 1E*).

To identify other immune-related genes that may cluster with SLAMF6, longitudinal RNA sequencing data were generated from CD4 T cells from two healthy human donors. Five groups of genes (clusters A-E) were identified (*Figure 1F*). Cluster A represents genes highly expressed in non-activated cells, and downregulated upon activation, such as *BCL2* and *JAK1*. Cluster B represents fast-rising genes undergoing transcription as early as 3–6 hr following activation, but down-regulated after that; genes in this cluster include the activation marker *CD69*. Clusters C and D include inter-mediate genes, upregulated after 6 hr (C) or between 12 and 24 hr (D), and downregulated later. Lastly, cluster E includes late-rising genes, such as *LAG3*. The *SLAMF6* transcript appears in cluster C, rising at 6 hr of activation and staying high after that (*Figure 1G*). Other genes in cluster C are *CD44*, encoding a glycoprotein that takes part in T cell activation (*Huet et al., 1989*), and *CD28* and *ICOS*, which encode co-stimulatory immune receptors (*Turka et al., 1990*; *Dong et al., 2001*). The increase in the transcription of receptors that are stably expressed at all times may hint at enhanced

recruitment and degradation of these receptors during activation, possibly in the immune synapse (*Onnis and Baldari, 2019*).

## SLAMF6 expressed in trans by a melanoma target inhibits antitumor T cell reactivity

Because SLAMF is constantly present on T cells, it is difficult to decipher its effect when it acts as a ligand, introduced in trans. To solve this problem, we generated a cell line derived from B16-F10/mhgp100 melanoma, over-expressing SLAMF6 (*Figure 2A*). Compared to the wild-type cell line, the SLAMF6-expressing melanoma cells co-cultured with Pmel-1 CD8+ T cells led to decreased IFN-γ secretion by the lymphocytes (*Figure 2B and C*). To evaluate the effect of SLAMF6 expression on melanoma rejection in vivo, SLAMF6-expressing melanoma was compared to the parental B16-F10/mhgp100 line, in an adoptive T cell transfer regimen, using activated Pmel-1 CD8+ T cells (*Figure 2D*). The B16-F10/mhgp100/SLAMF6$^+$ tumors grew more aggressively: on day 23, the mean tumor volume was 431 mm$^3$ in the SLAMF6-expressing melanomas, compared to 137 mm$^3$ in the non-modified tumors (p=0.04, Student's *t*-test) (*Figure 2E*).

These experiments show that trans-activation of SLAMF6 on lymphocytes, which in this system was achieved with the SLAMF6-expressing melanoma, inhibits the melanoma-specific CD8+ T cell response and allows rapid tumor growth.

## Establishment of Pmel-1 x SLAMF6 -/- mice as a source of SLAMF6-/- antigen specific lymphocytes

To evaluate the role of SLAMF6 in melanoma-cognate T cells, we generated a new mouse strain by breeding Pmel-1 mice with SLAMF6 -/- mice. The offspring of this cross represented a new strain,

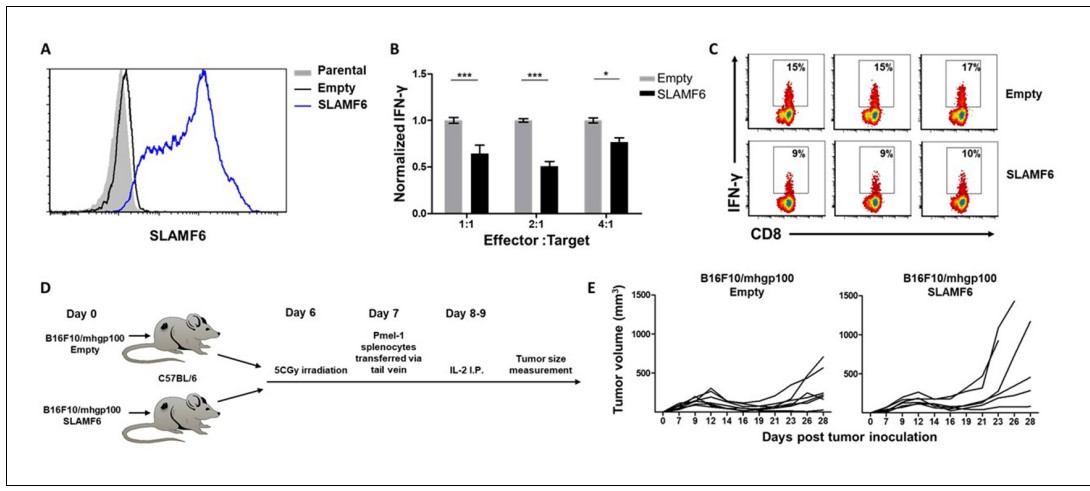

**Figure 2.** SLAMF6 expressed in trans by a melanoma target inhibits anti-tumor T cell reactivity. (**A**) SLAMF6 expression on B16-F10/mhgp100 parental or transfected (SLAMF6 or empty) melanoma cells. (**B**) Pmel-1 splenocytes were activated for 7 days with gp100$_{25-33}$ peptide and IL-2 (30 IU/ml), and then incubated overnight with B16-F10/mhgp100/empty or B16-F10/mhgp100/SLAMF6 melanoma cells at the indicated effector-to-target ratios. IFN-γ secretion was measured by ELISA. (**C**) Pmel-1 splenocytes were activated for 7 days with gp100$_{25-33}$ peptide and IL-2 (30 IU/ml), and then incubated overnight with B16-F10/mhgp100/empty or B16-F10/mhgp100/SLAMF6 melanoma cells. IFN-γ production was detected by intracellular staining and flow cytometry (gated on CD8+). Three replicates. The gating strategy is illustrated in *Figure 2—figure supplement 1*. (**D, E**) Pmel-1 splenocytes were expanded with gp100$_{25-33}$ peptide (1 µg/ml) and IL-2 (30 IU/ml) for 7 days. On day 7, cells were transferred i.v. into irradiated C57Bl/6 mice bearing palpable (1 week) B16-F10/mhgp100/empty or B16-F10/mhgp100/SLAMF6 tumors. IL-2 (0.25 × 10$^6$ IU) was administered i.p. twice a day for 2 days. Tumor growth was measured twice a week. Mice were sacrificed when the tumor reached 15 mm in diameter. (**D**) Scheme showing experimental layout. (**E**) Spider plot showing tumor volume [calculated as L (length) x W (width)$^2$ x 0.5]. One-way ANOVA. *, p<0.05, **, p<0.01, ***, p<0.001.

The online version of this article includes the following figure supplement(s) for figure 2:

**Figure supplement 1.** Gating strategy.

which could serve as a source of CD8+ T cells lacking SLAMF6 and expressing the transgenic TCR against the H-2D$^b$ gp100: 25–33 peptide (*Figure 3A*). Evaluation of the lymphocyte subsets in these mice showed a lower percentage (15%) of CD8+ cells in Pmel-1 x SLAMF6 -/- mouse spleens compared to Pmel-1 splenocytes (24%) (*Figure 3B* and *Figure 3—figure supplement 1A*). Despite the smaller percent of CD8+ cells in the spleens of the SLAMF6-deficient mice, the ratio of CD8+ subpopulations (naive, effector, effector memory, and central memory) was similar in both mouse strains (*Figure 3C*).

In the initial in vitro activation assays, it was already clear that the Pmel-1 x SLAMF6 -/- T cells have improved functional capacity. Their proliferative response to peptide stimulation was preserved, which was mandatory to produce ample numbers of CD8+ lymphocytes for the adoptive T cell transfer regimen (*Figure 3D*). CFSE dilution curves were identical in the two mouse strains, as were the activation-induced cell death (AICD) rates (*Figure 3—figure supplement 1B,C*). After 3 days of activation, the -/- mice had higher expression levels of CD25 and CD137 (4-1BB) activation markers (*Figure 3E*). In parallel, higher PD-1 expression was detected on day 7 (*Figure 3F*), which in this experimental context was initially taken as an indicator of activation, but was later attributed to SAP deficiency in the SLAMF6 -/- lymphocytes (see below in Results and Figure 5A and B). PD-1 overexpression was also noted by Lu et al. in iNKT cells from SFR -/- mice (*Lu et al., 2019*). The expression of other SLAM family members and ligands (CD48, LY9, CD244, CD8+4, CD319) on T cells during activation was similar in Pmel-1 and Pmel-1 x SLAMF6 -/- cells (*Figure 3—figure supplement 1D*).

Lastly, we phenotyped the T cells following 3-day and 7-day activation to assess subset ratios based on CD44 and CD62L differentiation markers. While initially, all Pmel-1 T cells were naive, only

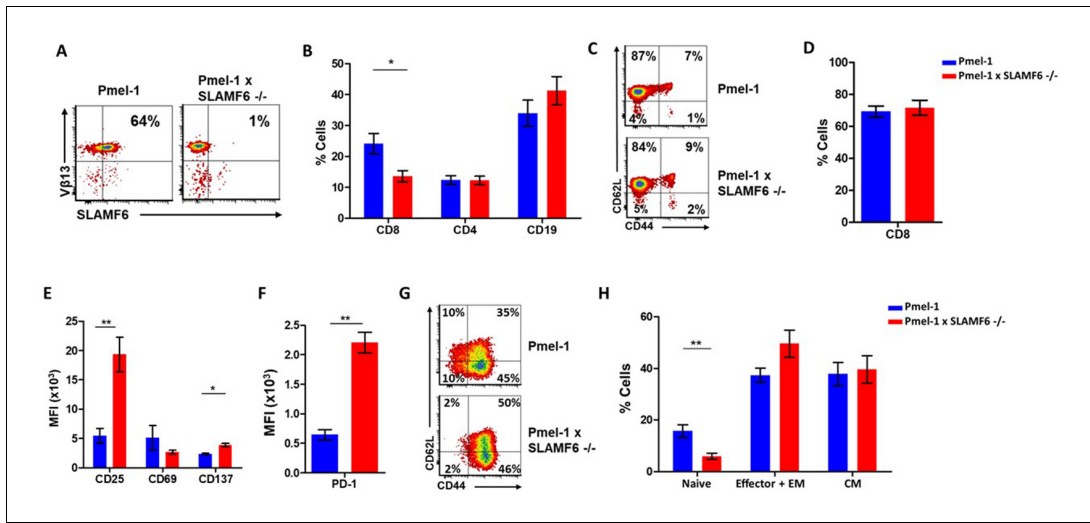

**Figure 3.** Establishment of Pmel-1 x SLAMF6 -/- mice as a source of SLAMF6-KO antigen-specific lymphocytes. (A) SLAMF6 and Vβ13 expression in Pmel-1 or Pmel-1 x SLAMF6 -/- splenocytes measured by flow cytometry. (B) Percent CD8+, CD4, and CD19 cells in spleens from Pmel-1 or Pmel-1 x SLAMF6 -/- untreated mice. (C) Pmel-1, and Pmel-1 x SLAMF6 -/- CD8+ untreated splenocytes were stained with anti-CD44 and anti-CD62L. One representative experiment is shown. (D) Percent CD8+ cells in Pmel-1 or Pmel-1 x SLAMF6 -/- splenocytes after 7 days of in vitro activation with gp100$_{25-33}$ peptide and IL-2 (30 IU/ml). (E) Flow cytometry for activation markers (CD25, CD69, CD137) in Pmel-1 or Pmel-1 x SLAMF6 -/- splenocytes after 3 days of in vitro activation, as in (D). Median fluorescence intensity (MFI) is shown. (F) Expression of PD-1 in Pmel-1 or Pmel-1 x SLAMF6 -/- CD8+ T cells after 7 days of in vitro activation, as in (D). Median fluorescence intensity (MFI) is shown. (G, H) After 7 days of activation, Pmel-1 and Pmel-1 x SLAMF6 -/- CD8+ T cells were stained with anti-CD44 and anti-CD62L. CD8+ subpopulations were defined for each mouse strain. (G) One representative experiment and (H) summary of subpopulations identified by flow cytometry in five experiments is shown. EM, effector memory, CM, central memory. Student t-test. *, p<0.05, **, p<0.01, ***, p<0.001.

The online version of this article includes the following figure supplement(s) for figure 3:

**Figure supplement 1.** Characterization of Pmel-1 x SLAMF6 -/- mice.

a negligible number of the activated Pmel-1 x SLAMF6 -/- cells remained in the naïve CD62L$^{high}$/CD44$^{low}$ state compared to 10% of the Pmel-1 cells. The complete shift of the activated Pmel-1 x SLAMF6 -/- lymphocytes towards effector and effector memory phenotypes indicates the strength of their response to activation (*Figure 3G and H*).

Pmel-1 x SLAMF6 -/- lymphocytes, generated to evaluate the effect of SLAMF6 on antigen-specific activated T cells, showed stronger activation and global acquisition of an effector phenotype. We attribute these results, in contrast to those previously obtained with lymphocytes from SLAMF6 -/- mice, to the role of SLAMF6 in the immune synapse, as the results could only be obtained if a synapse formation had been initiated via cognate TCRs.

## Pmel-1 X SLAMF6 -/- T cells have a better functional capacity

In the previous experiments, the effect of SLAMF6 deletion was evaluated in the activation and proliferation phases. To test if this superior activation also affects anti-tumor immunity, we characterized Pmel-1 x SLAMF6 -/- melanoma specific T cells in the effector phase.

Comparing IFN-γ secretion in response to melanoma cells by Pmel-1 versus Pmel-1 x SLAMF6 -/- lymphocytes showed that cytokine production by the Pmel-1 x SLAMF6 -/- lymphocytes was significantly higher at all effector-to-target ratios (p=0.05, *Figure 4A and B* and *Figure 4—figure supplement 1A and B*). In addition to IFN-γ, higher secretion of GM-CSF and lower levels of IL-10 and IL-13 were measured in the SLAMF6 -/- T cells (*Figure 4C and D*). Since GM-CSF is a strong recruiter of innate immune cells, while IL-10 and IL-13 drive suppressor traits, this secretion profile supports autocrine and paracrine immune activation. mRNA data validated the secretion assays (*Figure 4—figure supplement 1C*). Importantly, Pmel-1 x SLAMF6 -/- T cells produced higher levels of granzyme B in response to melanoma compared to Pmel-1 T cells, which are already strong killers due to their TCR design (*Figure 4E and F*). These results indicate that in the absence of the SLAMF6 modulatory effect, even strong cytolysis can be further enhanced.

To evaluate the antitumor activity of Pmel-1 x SLAMF6 -/- cells, we assessed adoptive cell transfer (ACT) of 7 day pre-activated gp100:25–33-specific, Pmel-1 or Pmel-1 x SLAMF6 -/- CD8+ T cells, transferred into mice bearing palpable B16-F10/mhgp100 melanoma in their back skin, followed by a 2-day course of intraperitoneal IL-2 (*Figure 4G–J*). The spider curves comparing melanoma growth following Pmel-1 versus Pmel-1 x SLAMF6 -/- T cell transfer revealed that in the first 4 weeks post-transfer, tumor growth was inhibited in both groups. However, on week 4, tumors treated with Pmel-1 ACT escaped control and grew again in six of the seven animals, whereas mice receiving Pmel-1 x SLAMF6 -/- ACT survived longer, and three of the seven treated mice remained tumor-free for over 80 days (*Figure 4H*). Vitiligo was noted in all mice attaining complete response, showing up earlier in the -/- group, at the 6th week, indicating the strength of the response (*Figure 4—figure supplement 1D*).

In a similar experiment, peptide-activated Pmel-1 CD8+ T cells or Pmel-1 x SLAMF6 -/- CD8+ T cells were transferred into mice bearing 7-day-old tumors. A week later, mice were sacrificed, and spleens, tumors, and tumor-draining lymph nodes were extracted and evaluated for the presence of transferred T cells, using flow cytometry to detect the gp100$_{25\text{-}33}$ tetramer. A higher proportion of gp100$_{25\text{-}33}$ tetramer+ cells was found in the draining lymph nodes of mice that had received Pmel-1 x SLAMF6 -/- cells (*Figure 4K*) compared to those who had received Pmel-1 cells. Tumors from both groups had a similar density of infiltrating lymphocytes (*Figure 4—figure supplement 1E*).

In summary, melanoma-specific T cells lacking SLAMF6 showed improved functional capacity both in vitro and in vivo and induced longer lasting tumor remission with longer tumor control compared to their wild-type counterparts.

## The contribution of *cis* and *trans* SLAMF6 interactions to CD8+ T cell function

SLAMF6 homotypic interactions may involve receptors on a same-cell population, in cis, or interactions in trans, between effector T cells and SLAMF6-expressing antigen-presenting targets. In *Figure 2*, we showed that aberrant presentation of SLAMF6 in trans on a melanoma target diminished T cell antitumor activity. Using the Pmel-1 x SLAMF6-/- CD8+ T cells, we then established matrices to determine the regulatory effect of SLAMF6 on the effector T cells and their antigen-presenting targets. SLAMF6-/- mice served as a source of APC devoid of SLAMF6, and peptide-loaded EL4 cells

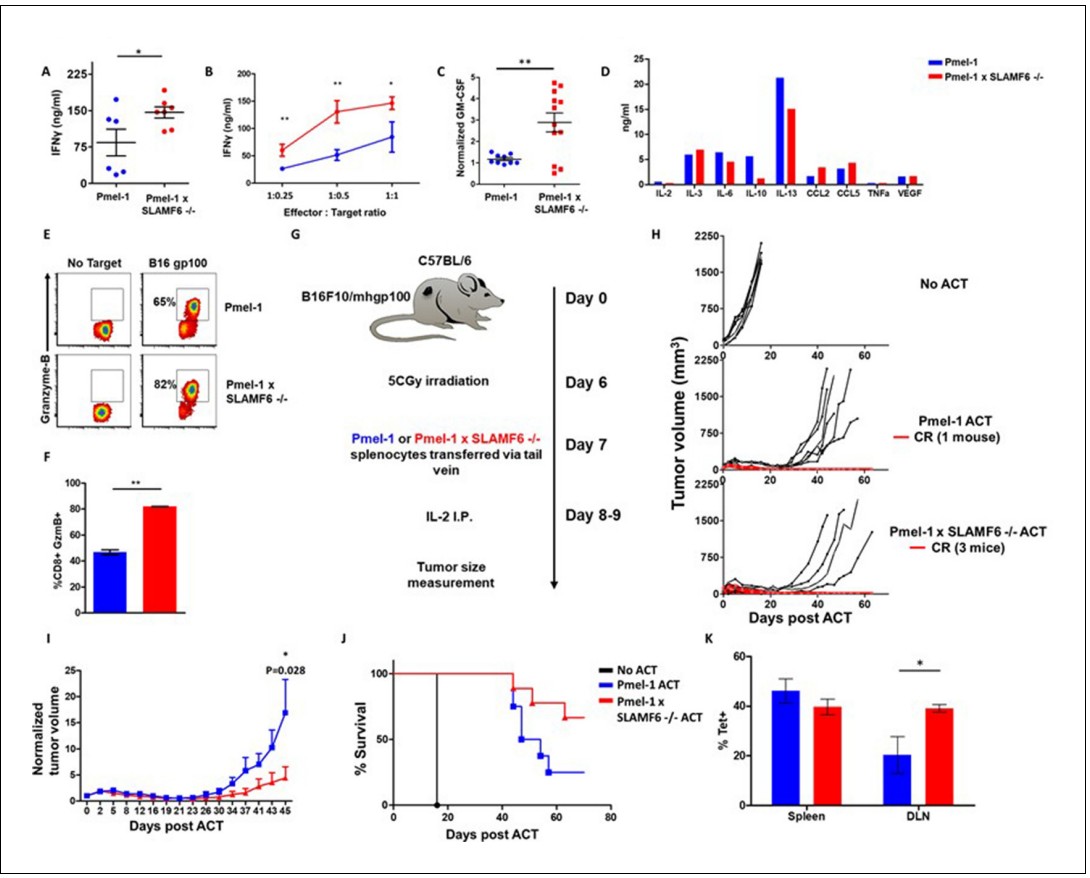

**Figure 4.** Pmel-1 x SLAMF6 -/- T cells have a better functional capacity. (**A–D**) Pmel-1 or Pmel-1 x SLAMF6 -/- splenocytes were activated for 7 days with gp100$_{25-33}$ peptide and IL-2 (30 IU/ml) and then incubated overnight with B16-F10/mhgp100 melanoma cells. (**A**) The cells were incubated at a 1:1 effector-to-target ratio. IFN-γ secretion was measured by ELISA. Each point represents one mouse. (**B**) The cells were incubated at the indicated effector-to-target ratios. IFN-γ secretion was measured by ELISA. (**C**) The cells were incubated at a 1:1 effector-to-target ratio. GM-CSF secretion was measured by ELISA. Each point represents one mouse. (**D**) Conditioned medium was collected and analyzed with Quantibody mouse cytokine array. (**E, F**) Pmel-1 or Pmel-1 x SLAMF6 -/- splenocytes were activated for 7 days with gp100$_{25-33}$ peptide and IL-2 (30 IU/ml) and then incubated for 16 hr with B16-F10/mhgp100 melanoma cells. Granzyme-B expression was detected by flow cytometry. One representative experiment (**E**) and a summary of triplicates (**F**) are shown. (**G–J**) B16-F10/mhgp100 mouse melanoma cells were injected s.c. into the back of C57BL/6 mice. Pmel-1 or Pmel-1 x SLAMF6 -/- mouse splenocytes were expanded with gp100$_{25-33}$ peptide in the presence of IL-2 (30 IU/ml). On day 7, Pmel-1 cells or Pmel-1 x SLAMF6 -/- cells were adoptively transferred i.v. into the irradiated tumor-bearing mice. N = 8 mice per group. Tumor size was measured three times a week. (**G**) Scheme of the experimental layout. (**H**) Spider plots showing tumor volume [calculated as L (length) x W (width)$^2$ x 0.5]. CR, complete response. (**I**) Normalized tumor volume (Mean ± SEM) until day 45, on which the first mouse had to be sacrificed. Tumor dimensions were normalized to the 1$^{st}$ measurement. (**J**) Kaplan Meier survival curve. (**K**) Percent T cells specific for gp100$_{25-33}$ peptide in the spleen or tumor draining lymph nodes (DLN) of mice sacrificed 7 days post-ACT. Tet, tetramer. Student t test. *, p<0.05, **, p<0.01, ***, p<0.001.

The online version of this article includes the following figure supplement(s) for figure 4:

**Figure supplement 1.** Additional results regarding Pmel-1 x SLAMF6 -/- T cell superiority.

transduced to express SLAMF6 served as another SLAMF6-expressing APC (*Figure 5B*). *Figure 5A* shows that knocking-out SLAMF6 in peptide-pulsed APCs enhances the response of Pmel-1 T cells, attesting to the inhibitory effect of the *trans* positioning of SLAMF6. A modest increase in function was observed in the SLAMF6 -/- T cells compared to the WT T cells when cultured with WT APCs, attesting to the improvement in the absence of *cis* positioned SLAMF6. However, when SLAMF6-/- T cells were activated by SLAMF6-/- APCs, their cytokine secretion increased even more (p=0.001,

**Figure 5.** The contribution of cis and trans SLAMF6 interactions to CD8+ T cell function. (**A**) Separated CD8+ splenocytes from Pmel-1 and Pmel-1 x SLAMF6 -/- were co-cultured ($1 \times 10^5$) overnight at the indicated ratios with non-T splenocytes from both mice splenocytes. IFN-$\gamma$ secretion was measured by ELISA. (**B**) SLAMF6 expression on EL4 parental or transfected (SLAMF6 or empty) cells. (**C**) Pmel-1 or Pmel-1 x SLAMF6 -/- splenocytes were activated for 7 days with gp100$_{25-33}$ peptide and IL-2 (30 IU/ml) and then incubated overnight with gp100$_{25-33}$ pulsed EL4 cells (empty or SLAMF6 transfected), at a 1:1 effector-to-target ratio. IFN-$\gamma$ secretion was measured by ELISA. IFN-$\gamma$ values are normalized to the results of EL4-empty for each mouse splenocytes. One-way ANOVA test. *, $p < 0.05$, **, $p < 0.01$, ***, $p < 0.001$.

Anova test). This finding suggests that not only the absence of SLAMF6 on the T cells improves their secretion capacity, but also SLAMF6-deficient APCs are inherently better. This observation was confirmed with the EL4 APCs (*Figure 5C*), which inhibited the WT Pmel-1 T cells, as expected, but surprisingly also inhibited the Pmel-1 x SLAMF6-/- CD8+ T cells, which were expected to be resistant to this effect.

These results show that the inhibitory effect of SLAMF6 derives from both in *cis* and in *trans* interactions. However, the role of SLAMF6 on APCs warrants further investigation.

## Mechanism associated with the inhibitory function of SLAMF6

The goal of the next series of experiments was to identify mechanisms underlying the improved effector function of Pmel-1 x SLAMF6 -/- lymphocytes. We initially evaluated the level of SAP, the primary adaptor required for SLAMF6 signaling, encoded by the *Sh2d1a* gene, which was intact in the SLAMF6 -/- mice (*Figure 6A*). SAP is a critical adaptor that recruits Fyn kinase to SLAMF6. However, while SAP transcript was found at similar levels in WT and SLAMF6 -/- lymphocytes, SAP protein was not detectable in SLAMF6 -/- cells (*Figure 6B*). The discrepancy between the transcript and the protein could be due to the rapid degradation of cytoplasmic SAP in its unbound form. SAP protein deficiency also implies that SLAMF6 is its major anchor in non-activated CD8+ T cells, even though SAP also mediates the inhibitory activity of PD-1 (*Peled et al., 2018*). As shown in *Figure 3F*, PD-1 expression is more than two-fold increased in 7 day activated Pmel-1 x SLAMF6 -/- lymphocytes compared to Pmel-1, but, as we have shown, this gap did not affect functionality. Thus, PD-1 overexpression can represent the failure of this receptor to generate a negative feedback loop in over-activated T cells.

Next, we measured by flow cytometry the level of phosphorylated ribosomal protein S6 (rpS6), an integrator of important signaling pathways, including PI3K/AKT/mTOR and RAS-ERK (*Figure 6C*). No difference in phosphorylated rpS6 was found in the Pmel-1 x SLAMF6 -/- T cells.

Therefore, we proceeded to identify transcription regulators whose activity differed between the Pmel-1 cells and the Pmel-1 x SLAMF6 -/- cells. The most prominent regulator found was T-bet, which increased more than twofold in activated Pmel-1 x SLAMF6 -/- splenocytes, followed by Eomes (*Figure 6D* and *Figure 6—figure supplement 1*). T-bet was originally described as the key transcription factor defining type 1 T helper (Th) cells; it has since been found to play a major role in the acquisition of effector functions by CD8+ T cells (*Szabo et al., 2000*). Type one inflammatory signals induce T-bet expression (*Joshi et al., 2007*), which is in line with the intensive IFN-$\gamma$ secretion we observed by the Pmel-1 x SLAMF6 -/- lymphocytes (*Figure 4A*).

Lastly, the level of immune receptors that mediate exhaustion was recorded during prolonged activation. Five days after the end of a 7-day activation course, Pmel-1 x SLAMF6 -/- T cells displayed

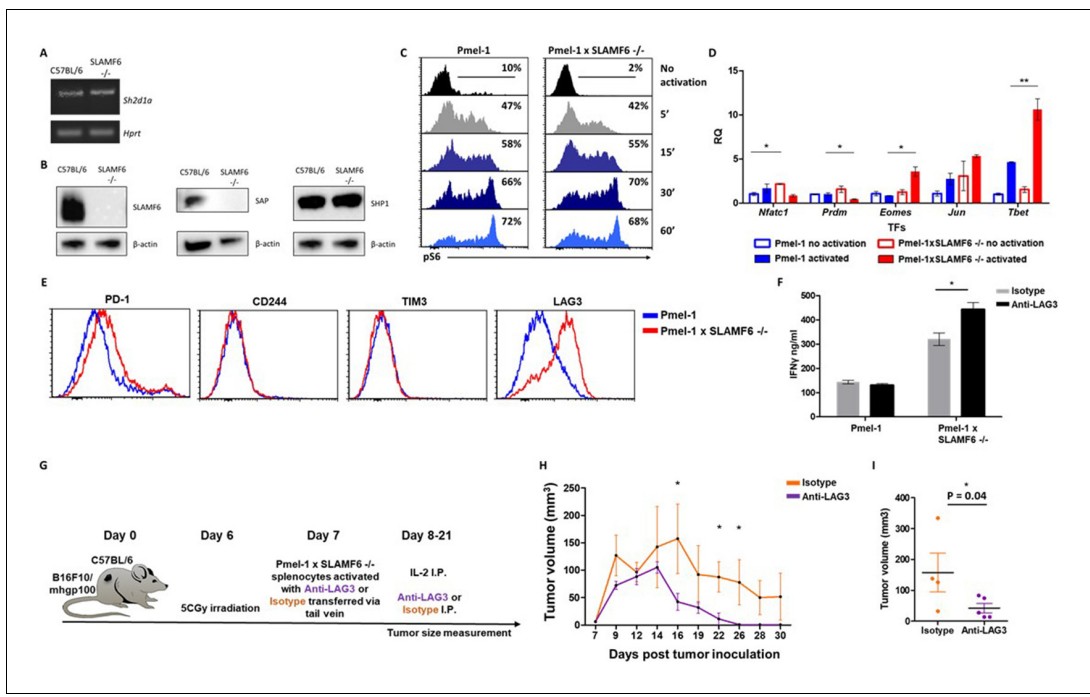

**Figure 6.** Mechanism associated with the inhibitory function of SLAMF6. (**A**) RNA expression of *Sh2d1a* transcript (SAP) in WT and SLAMF6 -/- splenocytes. (**B**) Immunoblot analysis of expression of SLAMF6, SAP and SHP-1 in WT and SLAMF6 -/- splenocytes. (**C**) Pmel-1 and Pmel-1 x SLAMF6 -/- splenocytes were activated with gp100$_{25-33}$ peptide for the indicated time points. At the end of the activation, cells were fixed and stained for phosphorylated S6. (**D**) Pmel-1 and Pmel-1 x SLAMF6 -/- splenocytes were either activated with gp100$_{25-33}$ peptide in the presence of IL-2 (30 IU/ml) for 18 hr or kept only with IL-2 for 18 hr (non-activated). After 18 hr, the cells were lysed, RNA was extracted, and quantitative RT-PCR for transcription factors expression was performed. Data was normalized to *Hprt* expression for each mouse strain. Values for each condition were normalized to Pmel-1 non-activated values for each gene. (**E**) Pmel-1 and Pmel-1 x SLAMF6 -/- splenocytes were expanded with gp100$_{25-33}$ peptide in the presence of IL-2 (30 IU/ml) for 7 days. After the expansion phase, the cells were kept for an additional 5 days without supplements. Expression of exhaustion markers was measured in Pmel-1 or Pmel-1 x SLAMF6 -/- splenocytes. (**F**) Pmel-1 and Pmel-1 x SLAMF6 -/- splenocytes were expanded with gp100$_{25-33}$ peptide in the presence of IL-2 (30 IU/ml) and 10 μg/ml anti-LAG-3 or isotype control for 7 days, and then incubated overnight with B16-F10/mhgp100 melanoma cells at a 1:1 effector-to-target ratio. IFN-γ secretion was measured by ELISA. (**G–I**) B16-F10/mhgp100 mouse melanoma cells were injected s.c. into the back of C57BL/6 mice. Pmel-1 x SLAMF6 -/- mouse splenocytes were expanded with gp100$_{25-33}$ peptide and IL-2 (30 IU/ml) in the presence of either Anti-Lag3 or Isotype control. On day 7, Isotype or Anti-Lag3 activated cells were adoptively transferred i.v. into the irradiated tumor-bearing mice. Anti-Lag3 or Isotype control were injected i.p. five times in the 2 weeks post-transfer. N = 5 mice per group. Tumor size was measured three times a week. (**G**) Scheme of the experimental layout. (**H**) Tumor volume (Mean ± SEM) until day 30 post-tumor inoculation. (**I**) Tumor volume on day 16 post-tumor inoculation. *, p<0.05, **, p<0.01.

The online version of this article includes the following figure supplement(s) for figure 6:

**Figure supplement 1.** Pmel-1 and Pmel-1 x SLAMF6 -/- splenocytes were either activated with gp100$_{25-33}$ peptide in the presence of IL-2 (30 IU/ml) for 18 hr or only kept with IL-2 (non-activated).

similar levels of PD-1, CD244 (SLAMF4), and TIM-3 to those in the Pmel-1 T cells, but higher expression of LAG-3 (*Figure 6E*). Hypothesizing that LAG-3 represents a compensatory mechanism for the enhanced activation of Pmel-1 x SLAMF6 -/- T cells, we used a blocking antibody against LAG-3 in the gp100 activation assay. We then measured IFN-γ secretion by activated T cells in response to B16-F10/mhgp100 melanoma cells. As expected, blocking LAG-3 on Pmel-1 x SLAMF6 -/- lymphocytes increased their cytokine secretion significantly, whereas it did not affect Pmel-1 cells. Overall, knocking out SLAMF6 together with LAG-3 blockade resulted in a three-fold increase in IFN-γ production (*Figure 6F*). To evaluate the combination of SLAMF6 -/- cells and LAG-3 blocking antibody, we conducted an ACT experiment in melanoma-bearing mice, using Pmel-1 x SLAMF6 -/- CD8+ T

cells activated in vitro in the presence of anti-LAG-3 and sustained by intraperitoneal IL-2 (days 8 and 9) and anti-LAG-3 (days 8, 10, 15, 18, 21). The control arm consisted of Pmel-1 x SLAMF6 -/- CD8+ T cells stimulated and sustained with an isotype antibody (*Figure 6G–I*). Mice treated by the combination of SLAMF6-deficient T cells and anti-LAG-3 antibody showed faster reduction and disappearance of their tumors (*Figure 6H*), apparent already on day 16 (p=0.04) (*Figure 6I*).

These results demonstrate that blocking the compensatory rise of LAG-3 on SLAMF6-/- T cells improved their anti-tumor effect even further.

## Discussion

The aim of this study was to characterize the role of SLAMF6 in CD8+ T cells, in the context of an antitumor response. The data obtained identify SLAMF6 as a receptor whose absence significantly improves CD8+-mediated tumor regression, suggesting that it is an inhibitory checkpoint.

Historically, SFRs were studied for their part in X-linked lymphoproliferative disease (XLP), a complex genetic immune dysfunction caused by a SAP mutation. XLP is characterized by a compromised immune response to Epstein-Barr virus (EBV) but also by unrestrained T lymphoblast proliferation, which is not necessarily EBV-induced. Thus, it is unclear whether loss of SAP converts all SFRs into 'super-inhibitory' receptors or whether, on the contrary, loss of SAP unleashes lymphocytes to proliferate, free from re-stimulation-induced apoptosis (*Katz et al., 2014*; *Kageyama et al., 2012*; *Zhao et al., 2012*; *Bottino et al., 2001*). Since SAP is an adaptor common to all SLAM family receptors, the role of each individual receptor was obscured by the shared defect. In this situation, SLAMF6 was considered a receptor with a dual function, depending on the interplay between SAP and SHP-1 and SHP-2, protein phosphatases that bind to tyrosines on the cytoplasmic tail of the receptor (*Veillette, 2010*; *Cannons et al., 2011*; *Detre et al., 2010*). SLAMF6 duality was echoed in data from Veillette that showed differing effects of SLAMF6 on NK cells, enhancing function in the priming phase while suppressing cells in the effector-phase (*Wu et al., 2016*). Also, mice lacking individual SFRs exhibit minor immune deviations (*Wu and Veillette, 2016*; *Cannons et al., 2011*; *Veillette, 2010*; *Calpe et al., 2008*).

In the past, we described that targeting SLAMF6 with its soluble ectodomain yielded CD8+ T cells that do not need IL-2 supplementation, either in vitro or in vivo, to eradicate established melanoma (*Eisenberg et al., 2018*). The beneficial effect of the soluble ectodomain of SLAMF6 prompted us to generate melanoma-specific SLAMF6 -/- T cells, to characterize the role of the receptor in a solid tumor model.

A key finding using the new Pmel-1 x SLAMF6 -/- mice described in this manuscript is the absence, in fact, of a dichotomy in SLAMF6 action in effector T cells. On the contrary, knocking-out SLAMF6 in murine antigen-specific CD8+ T cells disclosed an unequivocal inhibitory role for the receptor. In its absence, TCR triggering of anti-melanoma CD8+ T cells yielded a strong effector phenotype, higher IFN-γ secretion, improved cytolysis, and better outcomes in the adoptive transfer of SLAMF6 -/- anti melanoma CD8+ T cells to treat established melanoma. This study identifies SLAMF6 as a powerful inhibitor of antitumor immune response. The absence of viable SAP in SLAMF6 -/- lymphocytes hints that this adaptor takes a major part in the inhibitory effect of SLAMF6.

To explore the role of SLAMF6 in T cells without the confounding effects of its function in other cell types, we generated a system in which effector T cells interact with their tumor target based on specific epitope recognition and subsequently generate an immunological synapse. The synapse is a subcellular structure involved in the effect of SLAMF6 and is crucial for its study (*Zhao et al., 2012*). However, although we revealed the inhibitory effect of SLAMF6 in the Pmel-1 x SLAMF6 -/- mice, the source and configuration of SLAMF6/SLAMF6 homotypic binding in the wild-type situation were still difficult to characterize. We had to generate a SLAMF6-positive B16-F10/mhgp100 melanoma line to measure the effect, or more exactly, the degree of suppression, that SLAMF6 trans-activation has on the capacity of melanoma-cognate CD8+ T cells to eradicate tumors. As shown (*Figure 2E*), the SLAMF6-expressing melanoma suppressed T cell efficacy and consequently grew faster. This observation received further support from similar data generated with peptide-pulsed thymoma cells transduced to express SLAMF6 (*Figure 5C*). However, the improved IFNγ secretion of SLAMF6-/- T cells, when co-cultured with SLAMF6-lacking APCs compared to WT APCs, implies that an inherent

mechanism, most likely lack of *cis*-inhibition in the antigen presenting cells, is also responsible for the effect (*Figure 5A*).

The molecular mechanisms underlying the increased functional capacity of Pmel-1 T cells lacking SLAMF6 have common features with XLP, as the absence of SAP implies. But while XLP is a global defect of all cell types of the immune system, and therefore yields mixed derangements, the absence of SLAMF6 is remarkable for the enhanced functionality of CD8+ T cells, in which it is the dominating SFR.

The transcriptional landscape of SLAMF6 -/- T cells was governed by the higher expression of *T-bet*. T-bet is a transcription factor that contributes to Th1 and Th17 phenotypes in CD4 T cells. T-bet is prevalent in cytolytic innate lymphocytes residing in tissues and B cells of mouse strains prone to autoimmunity (*Plank et al., 2017*; *Nixon and Li, 2018*). The increased activity of T-bet in SLAMF6 -/- CD8+ T cells implies that T-bet-regulated pathways may operate in CD8+ T cells in the absence of functioning SLAMF6, generating 'type 1' inflammatory traits and high cytotoxicity. The improved production of IFN-γ and GM-CSF, in parallel with reduced IL-10 and IL-13, is also typical for type one phenotypes.

SLAMF6 should be distinguished from typical exhaustion markers because it is expressed on CD8 + T cells, regardless of their state of activation. Yigit et al. suggested that blocking SLAMF6 using an antibody can correct the exhaustion phase of T cells (*Yigit et al., 2019*), but we favor the notion that SLAMF6 hampers T cells at any stage, as reflected from the functional superiority of short-term activated Pmel-1 T cells. Depleting SLAMF6 improved CD8+ T cells in the short and long-term, as was most evident when the WT Pmel-1 cells induced the regression of melanoma only for a limited period while the Pmel-1 x SLAMF6 -/- cells led to lasting responses in mice (*Figure 4H*).

While searching for new immunotherapeutic targets, the field of immunotherapy is moving to combination therapies, and to biomarker-based treatment choices, to target the escape mechanisms used by tumors. From the results presented here, we conclude that SLAMF6 is an important checkpoint receptor with a significant inhibitory effect on T cells. The balance between SLAMF6 and LAG-3, and the enhancing effect of LAG3 blocking suggests that targeting both may have a valuable combinatorial, and perhaps even a synergistic, effect (*Figure 6G–I*).

In summary, we have shown that SLAMF6 is a constitutive inhibitory immune receptor; in its absence, CD8+ T cells acquire stronger reactivity against tumor cells. The strong effector trait is attributed to a series of T-bet-mediated transcriptional events that drive CD8+ T cells to exert strong cytotoxicity and achieve long-lasting tumor control. SLAMF6 is an attractive target for the development of checkpoint inhibitors for systemic treatment of cancer and for the improvement of antitumor cellular therapies.

## Materials and methods

### Key resources table

| Reagent type (species) or resource | Designation | Source or reference | Identifiers | Additional information |
|---|---|---|---|---|
| Strain, strain background *Mus musculus* Female) | C57BL/6 | Harlan laboratories | C57BL/6 | |
| Strain, strain background *Mus musculus* Female) | Pmel-1 | This paper | Pmel-1 | Kind gift from M. Baniyash |
| Strain, strain background *Mus musculus* Female) | SLAMF6 -/- | This paper | SLAMF6 -/- | Kind gift from I. Shachar |

*Continued on next page*

*Continued*

| Reagent type (species) or resource | Designation | Source or reference | Identifiers | Additional information |
|---|---|---|---|---|
| Strain, strain background *Mus musculus* Female) | Pmel-1 x SLAMF6 -/- | This paper | | Pmel-1 and SLAMF6-/- mice were bred to generate Pmel-1 X SLAMF6-/- mice according to the ethics requirements (Authority for biological and biomedical models, Hebrew University, Jerusalem, Israel). |
| Genetic reagent *Mus musculus* | SLAMF6 | SINO biological | HG11945-UT | pCMV3-mSLAMF6 |
| Cell line *Mus musculus* | B16-F10/mhgp100 | This paper | | Kind gift from Ken-ichi Hanada, Surgery Branch, NCI, NIH |
| Cell line *Mus musculus* | EL4 | This paper | | Kind gift from Lea Eisenbach, Weizmann Institute, Israel |
| Cell line (*Homo sapiens*) | TIL412 | This paper | | The cells were maintained in Lotem's laboratory |
| Biological sample (*Homo sapiens*) | PBMCs | This paper | | Blood drawn from donors recruited from the Boston community as part of the Phenogenetic Project and ImmVar Consortium |
| Antibody | Monoclonal Rat anti mouse CD16/32 (93) | Biolegend, San Diego, CA | 101302 | 0.2 µg/100 µl |
| Antibody | Monoclonal mouse anti mouse SLAMF6 (330-AJ) | Biolegend, San Diego, CA | 134610 | 0.2 µg/100 µl |
| Antibody | Monoclonal rat anti mouse TNFα (MP6-XT22) | Biolegend, San Diego, CA | 506314 | 0.2 µg/100 µl |
| Antibody | Monoclonal rat anti mouse CD19 (6D5) | Biolegend, San Diego, CA | 115521 | 0.2 µg/100 µl |
| Antibody | Monoclonal rat anti mouse CD44 (IM7) | Biolegend, San Diego, CA | 103016 | 0.2 µg/100 µl |
| Antibody | Monoclonal mouse anti mouse TIM3 (RMT3-23) | Biolegend, San Diego, CA | 119706 | 0.2 µg/100 µl |
| Antibody | Monoclonal rat anti mouse LAG3 (C9B7W) | Biolegend, San Diego, CA | 125210 | 0.2 µg/100 µl |
| Antibody | Monoclonal rat anti mouse CD3 (145–2 C11) | Biolegend, San Diego, CA | 100302 | 1 µg/ml |
| Antibody | Monoclonal mouse anti human CD3 (UCHT1) | BD Biosciences, San Jose, CA | 550368 | |
| Antibody | Monoclonal mouse anti human CD28 (CD28.2) | BD Biosciences, San Jose, CA | 556620 | |
| Antibody | Monoclonal rat anti mouse IFNγ (XMG1.2) | Biogems, Westlake Village, CA | 80812 | 0.2 µg/100 µl |
| Antibody | Monoclonal rat anti mouse CD8 (53–6.7) | Biogems, Westlake Village, CA | 10122 | 0.2 µg/100 µl |
| Antibody | Monoclonal rat anti mouse GZMB (NGZN) | Biogems, Westlake Village, CA | 72212 | 0.2 µg/100 µl |
| Antibody | Monoclonal rat anti mouse CD4 (GK1.5) | Biogems, Westlake Village, CA | 06112 | 0.2 µg/100 µl |

*Continued on next page*

*Continued*

| Reagent type (species) or resource | Designation | Source or reference | Identifiers | Additional information |
|---|---|---|---|---|
| Antibody | Monoclonal rat anti mouse CD25 (PC61.5) | Biogems, Westlake Village, CA | 07312 | 0.2 µg/100 µl |
| Antibody | Monoclonal rat anti mouse CD62L (MEL-14) | eBioscience, CA | 25-0621-81 | 0.2 µg/100 µl |
| Antibody | Monoclonal mouse anti mouse Vb13 (MR12-3) | eBioscience, CA | 17-5797-82 | 0.2 µg/100 µl |
| Antibody | Monoclonal Armenian hamster anti mouse CD69 (H1.2F3) | eBioscience, CA | 14-0691-82 | 0.2 µg/100 µl |
| Antibody | Monoclonal Armenian hamster anti mouse CD279 (J43) | eBioscience, CA | 12-9985-82 | 0.2 µg/100 µl |
| Antibody | Monoclonal rat anti mouse CD244 (eBio244F4) | eBioscience, CA | 14-2441-82 | 0.2 µg/100 µl |
| Antibody | Monoclonal Syrian hamster anti mouse CD137 (17B5) | eBioscience, CA | 12-1371-82 | 0.2 µg/100 µl |
| Antibody | Monoclonal anti human SLAMF6 (REA339) | Miltenyi Biotec, Bergisch Gladbach, Germany | Cd352 | 0.2 µg/100 µl |
| Antibody | Monoclonal rabbit anti pS6 (D57.2.2E) | Cell Signaling Technology, Danvers, MA | 4858 | 0.2 µg/100 µl |
| Antibody | Monoclonal rat anti mouse LAG-3 (C9B7W) | InVivoMab, BioXcell, NH | BE0174 | 10 µg/1 ml |
| Antibody | Monoclonal rat anti-Ly108 (3E11) | Merck, Kenilworth, NJ | MABF919 | 1:1000 |
| Antibody | Monoclonal rat anti-SAP (1A9) | Biolegend, San Diego, CA | 690702 | 1:1000 |
| Antibody | Monoclonal mouse anti b-actin (sc-47778) | Santa Cruz Biotechnology, TX | C4 | 1:1000 |
| Antibody | Monoclonal rabbit anti-SHP1 | This paper | | 1:1000 Generated in Andre' Veillette laboratory |
| Antibody | Monoclonal rabbit anti mouse CD4 (ab183685) | Abcam | EPR19514 | For immuno histochemistry |
| Antibody | Monoclonal rabbit anti mouse CD8+ (ab203035) | Abcam | ab203035 | For immuno histochemistry |
| Sequence-based reagent | Primers | | | All primers are listed in the primers table in the Materials and methods section |
| Peptide, recombinant protein | MART-1$_{26-35}$ | Biomer Technology, Cheshire, UK | | |
| Peptide, recombinant protein | gp100$_{25-33}$ | Genscript biotech, NJ | | |
| Commercial assay or kit | IFN-γ ELISA | Biolegend | 430801 | |
| Commercial assay or kit | GM-CSF ELISA | Biolegend | 432201 | |
| Commercial assay or kit | GenElute Mammalian Total RNA kit | Sigma Aldrich, MA | RTN70 | RNA production |

*Continued on next page*

*Continued*

| Reagent type (species) or resource | Designation | Source or reference | Identifiers | Additional information |
|---|---|---|---|---|
| Commercial assay or kit | qScript cDNA Synthesis kit | Quantabio, Beverly, MA | 95047 | RNA transformed to cDNA |
| Commercial assay or kit | RNeasy 96 kit | Qiagen, Hilden, Germany | 74181 | RNA production |
| Commercial assay or kit | Annexin V apoptosis detection kit | eBioscience | 88-8007-74 | Survival assay |
| Commercial assay or kit | Mouse CD8 T cell isolation kit | Stemcell technologies, Vancouver, CA | EasySep 19853A | CD8 isolation from total splenocytes |
| Commercial assay or kit | Quantibody mouse cytokine array | RayBiotech, Peachtree Corners, GA | QAM-CYT-1 | |
| Chemical compound, drug | IL-2 | Chiron, CA | recombinant human IL-2 | |
| Software, algorithm | FCS express five flow research edition | De Novo software | | |
| Software, algorithm | CellSens Entry 1.8 | Olympus Life Science | | Acquisition software for immunohistochemistry |
| Other | lyse/fix buffer | BD Biosciences | Cat: 558049 | |
| Other | PermII buffer | BD Biosciences | Cat: 558052 | |

## Plasmids

pCMV3-mSLAMF6 and pCMV3-negative control vectors were purchased from SINO Biological Inc, Eschborn, Germany.

## Cells

### Melanoma cells

Mouse melanoma B16-F10/mhgp100 cells (B16-F10 melanoma cells transduced with pMSGV1 retrovirus, which encodes a chimeric mouse gp100 with the human gp100$_{25-33}$ sequence) were a kind gift from Ken-ichi Hanada, Surgery Branch, NCI, NIH. The cells were cultured in RPMI 1640 supplemented with 10% heat-inactivated fetal calf serum (FCS), 2 mmol/L L-glutamine, and combined antibiotics (all from ThermoFisher Scientific, MA). Lines were regularly tested and were mycoplasma free.

### EL4 cells

EL4 cells, a thymoma cell line derived from a C57BL/6 (H-2b) mouse, were a kind gift from Lea Eisenbach, Weizmann Institute, Israel. The cells were cultured in RPMI 1640 supplemented with 10% heat-inactivated fetal calf serum (FCS), 2 mmol/L L-glutamine, and combined antibiotics (all from Thermo-Fisher Scientific, MA). Lines were regularly tested and were mycoplasma free.

### Aberrant LY108 expression on melanoma and EL4 cells

*The* B16-F10/mhgp100 murine melanoma cell line and EL4 cells were transfected with pCMV3—hygromycin-mSLAMF6 using lipofectamine 2000 (ThermoFisher). Hygromycin-resistant cells were sub-cloned, and the stably transfected cells were labeled with anti-LY108 Ab and sorted in an ARIA-III sorter. B16-F10/mhgp100 cells and EL4 cells transfected with the empty vector were sub-cloned using hygromycin resistance.

### Tumor-infiltrating lymphocytes (TILs)

Fresh tumor specimens taken from resected metastases of melanoma patients were used to release TILs using a microculture assay (*Lotem et al., 2002*). Human lymphocytes were cultured in complete medium (CM) consisting of RPMI 1640 supplemented with 10% heat-inactivated human AB serum, 2 mmol/l L-glutamine, 1 mmol/l sodium pyruvate, 1% nonessential amino acids, 25 mmol/l HEPES (pH

7.4), 50 µmol/l 2-ME, and combined antibiotics (all from ThermoFisher). CM was supplemented with 6000 IU/ml recombinant human IL-2 (rhIL-2, Chiron, CA).

## Cloning of peptide-specific TILs

On day 14 after TIL initiation, lymphocytes were washed with PBS, re-suspended in PBS supplemented with 0.5% bovine serum albumin (BSA), and stained with FITC-conjugated HLA-A*0201/MART-1$_{26-35}$ dextramer (Immudex, Copenhagen Denmark) for 30 min at 4°C. Lymphocytes were then incubated with allophycocyanin-conjugated mouse anti-human CD8+ (eBioscience, CA) for an additional 30 min at 4°C and washed. CD8+ lymphocytes, positively stained with the dextramer (CD8+dextramer$^+$ cells), were sorted with a BD Biosciences FACS Aria and directly cloned at one or two cells per well in 96-well plates in the presence of 30 ng/ml anti CD3 (eBioscience), 6000 IU/ml rhIL-2, and 4 Gy-irradiated $5 \times 10^4$ allogeneic PBMCs as feeder cells. Five days later, 6000 IU/ml rhIL-2 was added and renewed every 2 days after that. On day 14, the clones were assayed for IFN-γ secretion in a peptide-specific manner following their co-incubation with T2 cells pulsed with MART-1$_{26-35}$ (commercially synthesized and purified [>95%] by reverse-phase HPLC by Biomer Technology, Cheshire, UK) by ELISA (R and D Systems, Minneapolis, MN). The MART-1$_{26-35}$–-reactive clones were further expanded in a second-round of exposure to 30 ng/ml anti-CD3, and 6000 IU/ml rhIL-2 in the presence of 50-fold excess irradiated feeder cells.

## Antibodies

For flow cytometry, cells were labeled with the following reagents: anti-CD16/32 (93), anti-SLAMF6 (330-AJ), anti-TNFa (MP6-XT22), anti-CD19 (6D5), anti-CD44 (IM7), anti-TIM3 (RMT3-23), and anti-LAG-3 (C9B7W) (all from Biolegend, San Diego, CA). Anti-IFN-γ (XMG1.2), anti-CD8+ (53–6.7), anti-GZMB (NGZN), anti-CD4 (GK1.5), and anti-CD25 (PC61.5) were from Biogems. Anti-CD62L (MEL-14), anti-Vb13 (MR12-3), anti-CD69 (H1.2F3), anti-CD137 (17B5), anti-PD1 (J43), and anti-CD244 (eBio244F4) were from eBioscience. Anti-human SLAMF6 (REA) was purchased from Miltenyi Biotec, Bergisch Gladbach, Germany. Anti-pS6 (D57.2.2E) was from Cell Signaling Technology, Danvers, MA.

Anti-LAG-3 (C9B7W) and the corresponding isotype were from InVivoMab, BioXcell, NH, USA. The Mart-1$_{26-35}$ iTag MHC tetramer was from MBL, Woburn, MA.

For Immunobloting: anti-Ly108 (Rat, 3E11, Merck, Kenilworth, NJ), anti-SAP (Rat, 1A9, Biolegend), anti-β actin (Mouse, sc-47778, Santa Cruz Biotechnology, TX), anti-SHP1 (Rabbit, generated in A.V.'s laboratory).

## Mice

C57BL/6 mice were purchased from Harlan laboratories. Pmel-1 (a kind gift from M. Baniyash) and SLAMF6 -/- mice (a kind gift from I. Shachar) were self-bred.

The Pmel-1 mice carry a rearranged TCR specific for a 9-mer epitope (25-32) from murine Pmel 17, overexpressed on transformed melanocytes and homologous to the human melanoma-associated antigen gp100.

All experiments were performed with 8- to 12-week-old female mice.

*Generation of Pmel-1 x SLAMF6 -/- mice.* Pmel-1 and SLAMF6-/- mice were bred to generate Pmel-1 X SLAMF6-/- mice according to the ethics requirements (Authority for biological and biomedical models, Hebrew University, Jerusalem, Israel).

When the mice reached 3 weeks of age, 2 mm of the mouse tail were cut, 200 µl 50 mM NaOH 0.2 Mm EDTA were added, and the tails were incubated at 95°C for 20 min for DNA purification. 200 µl 80 mM TRIS-HCL, pH5, were added to stop the reaction. The DNA purified from the tails was used in PCR reactions for genotyping of mice in the SLAMF6 locus on chromosome 1 (primers adapted from the Jackson laboratories website) and in the Pmel-1 locus on chromosome 2 (*Ji et al., 2014*). The identification of the genomic insertion site of the Pmel-1 TCR α and β transgenes was performed by next-generation sequencing.

## Splenocyte activation

Pmel-1 or Pmel-1xSLAMF6-/- mouse splenocytes ($2 \times 10^6$/ml) were activated with 1 µg/ml of mouse gp100$_{25-33}$ peptide for 6 days with IL-2 30 IU/ml. Fresh medium containing IL-2 was added every other day.

## In vitro assays

*RNA isolation and qPCR.* RNA was isolated from cells using the GenElute Mammalian Total RNA kit (Sigma Aldrich, MA) according to the manufacturer's protocol. RNA was then transcribed to cDNA using qScript cDNA Synthesis kit (Quantabio, Beverly, MA) according to the manufacturer's instructions, and RT-PCT or qRT-PCR was performed using the following primers:

| Gene | Forward primer | Reverse primer |
|------|---------------|----------------|
| Human SLAMF6 | CTGTTCCAATCGCTCCTGTT | GGGGTTAAGCTGCTTTGTGA |
| Human HPRT | GAGGATTTGGAAAGGGTGTTT | CATCTCGAGCAAGACGTTCA |
| Mouse Sh2d1a | AAAATCAGCAGGGAGACCG | TCAACTGGATACTGCAGAGG |
| Mouse IL10 | GGACAACATACTGCTAACCGAC | CACTTCTACCAGGTAAAACTGG |
| Mouse IL13 | TTAAGGAGCTTATTGAGGA | GTCCACACTCCATACCAT |
| Mouse Blimp1 | TTCAAGCCGAGGCATCCTTA | CCTTCGGTATGTACTCCTTAC |
| Mouse Eomes | ACCGGCACCAAACTGAGA | AAGCTCAAGAAAGGAAACATGC |
| Mouse Fos | CGGGTTTCAACGCCGACTA | TTGGCACTAGAGACGGACAGA |
| Mouse Jun | CCAGAAGATGGTGTGGTGTTT | CTGACCCTCTCCCCTTGC |
| Mouse Tbx21 | AGGGGGCTTCCAACAATG | AGACGTGTGTGTTAGAAGCACTG |
| Mouse TCF7 | CAAGGCAGAGAAGGAGGCTAAG | GGCAGCGCTCTCCTTGAG |
| Mouse NFATc1 | CAGTTATGTGTCCCCTAGTGT | GGATGATTGGCTGAAGGAA |
| Mouse PRDM | AAGTTTCAAGGACTGGCAGA | GGTGGTCGTTCACTATGTAT |
| Mouse Egr2 | TTTGACCAGATGAACGGAGT | CTGGTTTCTAGGTGCAGAGA |
| Mouse Batf | AAGAGCCGACAGAGACAGA | TTTCTCCAGGTCCTCACTCT |
| Mouse Gata3 | AACATCGATGGTCAAGGCAACC | GTGGGTCGGAGGATACCTCT |
| Mouse Foxp3 | TACAGTGCCCCTAGTCATGGT | ATGGGCATCCACAGTGGAG |
| Mouse HPRT | GCCGAGGATTTGGAAAAAGTG | GCCTCCCATCTCCTTCATGAC |

RT-PCR for *Sh2d1a* was performed in the SensQuest lab cycler machine (Danyel Biotech); the products were then run on 1.5% agarose gel.

## Longitudinal expression of SLAMF6

TIL412 was activated using plate-bound anti-CD3 1 µg/ml supplemented with IL-2. On every other day, $1 \times 10^5$ cells were stained in triplicates to evaluate SLAMF6 expression using anti-SLAMF6 (Miltenyi Biotec) or the corresponding isotype control. Pmel-1 splenocytes were activated using 1 µg/ml mouse gp100$_{25-33}$ peptide supplemented with 30 IU/ml IL-2 for 6 days, and SLAMF6 expression was tested at the indicated time points using anti-SLAMF6 (Biolegend, clone 330-AJ).

## RNA sequencing of activated CD4 cells

Human PBMCs were isolated by Ficoll-Paque (GE Healthcare Life Sciences, Pittsburgh, PA) centrifugation from blood drawn from donors recruited from the Boston community as part of the Phenogenetic Project and ImmVar Consortium. Naive CD4 T cells were isolated from PBMCs by negative selection using the human Naive CD4$^+$ T Cell Isolation Kit II (Miltenyi Biotec). The isolation was >97% pure as assessed by flow cytometry. For each condition, 50,000 T cells were used. Beads used for cell stimulation were generated by incubating $4 \times 10^6$ CELLection Pan Mouse IgG beads (ThermoFisher) with antibodies (BD Biosciences, San Jose, CA) against CD3 (UCHT1; 10.67 ng) and CD28 (CD28.2; 5.33 ng) complemented to 20 ng total protein with control IgG1 (MOPC-31C) for CD3/CD28 stimulation at the following time points: 0, 3, 6, 12, 24, and 72 hr. RNA for sequencing was isolated using an RNeasy 96 kit (Qiagen, Hilden, Germany), treated with DNase I (New England

Biolabs, Ipswich, MA), and converted to sequencing libraries using the Smartseq2 protocol (*Picelli et al., 2014*). Libraries were sequenced on a Novaseq S2 (Illumina, San Diego, CA) with paired-end 50 bp reads. (Donors ID: Donor A – IGTB3, Donor B – IGTB968).

## RNA sequencing results analysis

We used Bowtie v1.0.0 to align raw data to the UCSC hg19 transcriptome and RSEM v1.2.8 to quantify gene expression levels (TPM – transcripts per million). Immune-related genes were selected from the RNA sequencing results based on a unified list of genes created from the Immunogenetic Related Information Source (IRIS) list and the MAPK/NFKB Network list (https://www.innatedb.com/redirect.do?go=resourcesGeneLists).

For the immune-related genes, heatmaps were generated using the Morpheus online tool (https://software.broadinstitute.org/morpheus/).

## Intracellular cytokine staining

Mouse splenocytes ($1 \times 10^5$) were co-cultured for 6 hr at 37°C at a 1:1 ratio with the indicated target melanoma cells or activated with 1 µg/ml anti-CD3 (Biolegend, clone: 145–2 C11). After 2 hr, Brefeldin A (Biolegend) was added (1:1000). After incubation the cells were washed twice with PBS and stained with anti-CD8+ (Biogems) for 30 min at room temperature. Following fixation and permeabilization (eBioscience protocol), intracellular IFN-γ and TNF-α were labeled with anti- IFN-γ and anti-TNF-α antibodies, respectively (TNF-α, clone: MP6-XT22, Biolegend, and IFN-γ, clone: XMG1.2, Biogems) for 30 min at room temperature. Cells were washed with permeabilization buffer, resuspended in FACS buffer, and subjected to flow cytometry.

## Intracellular staining for phosphorylated proteins

For intracellular staining of phosphorylated proteins, cells were activated for different times, fixed with lyse/fix buffer (BD Biosciences), permeabilized with Perm II buffer (BD Biosciences) and stained with fluorescence-labeled antibodies against pS6 (Cell Signaling, D57.2.2E) in stain buffer (BD Biosciences).

## Cell viability assay

Following expansion, splenocytes were washed, and $1 \times 10^5$ cells were cultured in CM. After 7 days, cells were washed and labeled with the Annexin V apoptosis detection kit (eBioscience), according to the manufacturer's instructions. Cells were analyzed by flow cytometry.

## Proliferation assay

Fresh splenocytes were labeled with CFSE (*Quah et al., 2007*) and activated as described above. At the indicated days, cells were counted, labeled with anti-CD8+ Ab (Biogems, clone: 53–6.7), and subjected to flow cytometry.

## Co-culture of T cells with splenocytes

Mouse CD8+ T cells were separated using Mouse CD8+ T cell isolation kit (EasySep 19853A, Stemcell Technologies, Vancouver, CA) and $1 \times 10^5$ cells were co-cultured overnight at the indicated ratios with non-T-splenocytes. Conditioned medium was collected, and mouse IFN-γ secretion was detected by ELISA (Biolegend).

## Interferon-gamma *secretion*

Mouse splenocytes, previously activated for 7 days, were co-cultured ($1 \times 10^5$) overnight at a 1:1 ratio with the indicated target cells or activated with 1 µg/ml plate-bound anti -CD3 (Biolegend, clone: 145–2 C11) as indicated in each experiment. Conditioned medium was collected, and mouse IFN-γ secretion was detected by ELISA (Biolegend).

## GM-CSF secretion

Mouse splenocytes, previously activated for 7 days, were co-cultured ($1 \times 10^5$) overnight at a 1:1 ratio with the indicated target melanoma cells. Conditioned medium was collected, and mouse GM-CSF secretion was measured by ELISA (Biolegend).

## Immunohistochemistry

For histological analysis, spleen and tumor tissues were cut into 5 µm sections, deparaffinized with xylene, and hydrated with graded ethanol. Endogenous peroxidase was blocked using 3% $H_2O_2$ for 5 min. and 0.01M citrate buffer (pH 6.0) was used for antigen retrieval. The samples were cooked in a pressure cooker at maximum power for 13 min and then at 40% power for an additional 15 min, and left to cool for 30 min until they reached room temperature. All slides were then washed in PBS and blocked for 30 min with CAS block at room temperature. The tissues were incubated with rabbit anti-CD4 Ab (ab183685) or rabbit anti-CD8+ Ab (ab203035) diluted in CAS-Block (1:1000 or 1:250, respectively) overnight at 4°C. The following day, the slides were washed in PBS, incubated with Dako anti-rabbit secondary antibody for 30 min and developed with AEC for 10 min. After several washes in PBS and dH$_2$O, the slides were counterstained with hematoxylin, rinsed in H$_2$O, and covered with fluoromount (ThermoFisher) and a cover-slip. The photos were taken with an Olympus BX50 microscope and Olympus DP73 camera at room temperature. The acquisition software used was CellSens Entry 1.8.

## Killing assay

Splenocytes were co-cultured for 19 hr at 37°C at a 1:1 ratio with the indicated target melanoma cells. After 16 hr, Brefeldin A (Biolegend) was added (1:1000). At the end of the incubation, the cells were washed twice with PBS and stained with aCD8+ (Biogems, clone 53–6.7) for 30 min at room temperature. Following fixation and permeabilization (eBioscience protocol), intracellular granzyme-B was labeled with anti-GzmB antibody (Biogems, clone: NGZB) for 30 min at RT. Cells were washed with permeabilization buffer, resuspended in FACS buffer, and subjected to flow cytometry.

## Cytokine array

Day seven activated mouse splenocytes were co-cultured ($1 \times 10^5$) overnight at a 2:1 ratio with the B16-F10/mhgp100 melanoma cells. Conditioned medium was collected and used in the Quantibody mouse cytokine array (QAM-CYT-1, RayBiotech, Peachtree Corners, GA).

## Immunobloting

Cells were lysed using RIPA buffer, and protein concentrations were tested using Bradford quantification. Equal concentrations of lysates were resuspended in SDS sample buffer (250 mM Tris-HCl [pH 6.8], 5% w/v SDS, 50% glycerol, and 0.06% w/v bromophenol blue) for 5 min at 95°C. Proteins were separated by SDS PAGE and transferred to a PVDF membrane. Membranes, blocked with 1% milk solution, were incubated with primary antibodies overnight at 4°C, followed by incubation with HRP-conjugated secondary antibodies for 1 hr at RT (Jackson ImmunoResearch Laboratories). Signals were detected by enhanced chemiluminescence reagents (Clarity Western ECL Substrate, Bio-Rad, Hercules, CA).

## Flow cytometry

After blocking Fc receptors with anti-CD16/CD32 antibody, cells were stained with antibodies or tetramers on ice or at room temperature for 25 min, according to the manufacturer's instructions.

Subsequently, cells were washed and analyzed using CytoFlex (Beckman Coulter, CA), and flow cytometry-based sorting was done in an ARIA-III sorter. Flow cytometry analysis was done using FCS express five flow research edition (De Novo software).

## In vivo assays

Adoptive cell transfer experiments:

a. B16-F10/mhgp100-empty vector or B16-F10/mhgp100-SLAMF6 transfected mouse melanoma cells ($0.5 \times 10^6$) were injected s.c. into the back of C57BL/6 mice. Pmel-1 mouse splenocytes

$(2 \times 10^6/\text{ml})$ were expanded with 1 µg/ml of mouse gp100$_{25\text{-}33}$ peptide in the presence of IL-2 (30 IU/ml). Fresh medium containing IL-2 was added every other day. On day 7, $10^7$ Pmel-1 cells were adoptively transferred i.v. into 500 CGy-irradiated tumor-bearing mice. $0.25 \times 10^6$ IU/100 µl IL-2 was administered i.p. twice daily for 2 days. Tumor size and mouse weight were measured three times a week. Follow-up was conducted until the ethical humane endpoint was reached.

b. B16-F10/mhgp100 mouse melanoma cells $(0.5 \times 10^6)$ were injected s.c. into the back of C57BL/6 mice. Pmel-1 or Pmel-1xSLAMF6 -/- mouse splenocytes $(2 \times 10^6/\text{ml})$ were expanded with 1 µg/ml of gp100$_{25\text{-}33}$ peptide in the presence of IL-2 (30 IU/ml). Fresh medium containing IL-2 was added every other day. On day 7, $10^7$ Pmel-1 cells or Pmel-1xSLAMF6 -/- cells were adoptively transferred i.v. into 500 CGy-irradiated tumor-bearing mice. $0.25 \times 10^6$ IU/100 µl IL-2 was administered i.p. twice a day for 2 days. Tumor size and mouse weight were measured three times a week. Follow up was conducted until the ethical humane endpoint was reached or until day 80 (for mice that showed complete tumor remission).

c. The experiment was performed as in (b). One week after ACT, mice were sacrificed, and tumors, DLN, and spleens were harvested for further analysis by flow cytometry or immunohistochemistry.

d. B16-F10/mhgp100 mouse melanoma cells $(0.5 \times 10^6)$ were injected s.c. into the back of C57BL/6 mice. Pmel-1 x SLAMF6 -/- mouse splenocytes $(2 \times 10^6/\text{ml})$ were expanded with 1 µg/ml of the gp100$_{25\text{-}33}$ peptide in the presence of IL-2 (30 IU/ml) and either anti-LAG-3 Ab (10 µg/ml) or the corresponding isotype. Fresh medium containing IL-2 and the Ab was added every other day. On day 7, $10^7$ Pmel-1 x SLAMF6 -/- cells activated in the presence of anti-LAG-3 or isotype were adoptively transferred i.v. into 500 CGy-irradiated tumor-bearing mice. $0.25 \times 10^6$ IU/100 µl IL-2 was administered i.p. twice a day for 2 days. 100 µg anti-LAG-3 or isotype was administered i.p. in 100 µl PBS five times during the 2 weeks post-adoptive transfer. Tumor size and mouse weight were measured three times a week. Follow up was conducted until day 30.

Animal studies were approved by the Institutional Review Board - Authority for biological and biomedical models, Hebrew University, Jerusalem, Israel (MD-14602–5 and MD-15421–5).

*Statistics.* Statistical significance was determined by unpaired t-test (two-tailed with equal SD) using Prism software (GraphPad). A p-value<0.05 was considered statistically significant. Analysis of more than two groups was performed using the one-way ANOVA test. *, $p \leq 0.05$; **, $p \leq 0.01$; ***, $p \leq 0.001$. For each experiment, the number of replicates performed and the statistical test used are stated in the corresponding figure legend.

## Acknowledgements

The authors wish to acknowledge the devoted technical work of Anna Kuznetz and Yael Gelfand. We thank Eli Pikarsky and Ofer Mandelboim for helpful discussions, and Karen Pepper for editing the manuscript.

We thank Kathleen B Yates and W Nicholas Haining, who provided training and their expertise towards the design of a range of beads to optimize control of the degree and nature of antigenic stimulation of T cells in vitro.

We thank the subjects in the PhenoGenetic Project for donating the blood used in our study. This project was supported by NIH/NCI R01CA208756 (PI: NH)

## Additional information

### Funding

| Funder | Grant reference number | Author |
| --- | --- | --- |
| Dr. Miriam and Sheldon G. Adelson Medical Research Foundation | | Emma Hajaj<br>Galit Eisenberg<br>Shiri Klein<br>Shoshana Frankenburg<br>Sharon Merims<br>Inna Ben David<br>Jonathan E Cohen<br>Michal Lotem |

| Melanoma Research Alliance | Emma Hajaj<br>Galit Eisenberg<br>Shiri Klein<br>Shoshana Frankenburg<br>Sharon Merims<br>Inna Ben David<br>Jonathan E Cohen<br>Michal Lotem |
|---|---|
| Canadian Institutes of Health Research | Emma Hajaj<br>Galit Eisenberg<br>Shiri Klein<br>Shoshana Frankenburg<br>Sharon Merims<br>Inna Ben David<br>Jonathan E Cohen<br>Michal Lotem |
| International Development Research Centre | Emma Hajaj<br>Galit Eisenberg<br>Shiri Klein<br>Shoshana Frankenburg<br>Sharon Merims<br>Inna Ben David<br>Jonathan E Cohen<br>Michal Lotem |
| Israel Science Foundation | Emma Hajaj<br>Galit Eisenberg<br>Shiri Klein<br>Shoshana Frankenburg<br>Sharon Merims<br>Inna Ben David<br>Jonathan E Cohen<br>Michal Lotem |
| Azrieli Foundation | Emma Hajaj<br>Galit Eisenberg<br>Shiri Klein<br>Shoshana Frankenburg<br>Sharon Merims<br>Inna Ben David<br>Jonathan E Cohen<br>Michal Lotem |
| Deutsche Forschungsgemeinschaft | Emma Hajaj<br>Galit Eisenberg<br>Shiri Klein<br>Shoshana Frankenburg<br>Sharon Merims<br>Inna Ben David<br>Jonathan E Cohen<br>Michal Lotem |
| Rosetrees Trust | Emma Hajaj<br>Galit Eisenberg<br>Shiri Klein<br>Shoshana Frankenburg<br>Sharon Merims<br>Inna Ben David<br>Jonathan E Cohen<br>Michal Lotem |
| Perlstein Family Fund | Emma Hajaj<br>Galit Eisenberg<br>Shiri Klein<br>Shoshana Frankenburg<br>Sharon Merims<br>Inna Ben David<br>Jonathan E Cohen<br>Michal Lotem |
| Fred Lovejoy Resident Research Fund Awards | Sarah E Henrickson |

| International Development Research Centre | 108403 | Andre Veillette |
| Canadian Institutes of Health Research | FDN-143338 | Andre Veillette |
| National Cancer Institute | R01CA208756 | Nir Hacohen |

The funders had no role in study design, data collection and interpretation, or the decision to submit the work for publication.

## Author contributions

Emma Hajaj, Conceptualization, Data curation, Formal analysis, Investigation, Methodology, Writing - original draft, Project administration, Writing - review and editing; Galit Eisenberg, Conceptualization, Data curation, Investigation, Methodology, Project administration, Writing - review and editing; Shiri Klein, Conceptualization, Data curation, Writing - review and editing; Shoshana Frankenburg, Conceptualization, Writing - original draft, Writing - review and editing; Sharon Merims, Jonathan E Cohen, Conceptualization, Writing - review and editing; Inna Ben David, Data curation, Writing - review and editing; Thomas Eisenhaure, Data curation, Software, Formal analysis, Writing - original draft; Sarah E Henrickson, Conceptualization, Data curation, Writing - original draft; Alexandra Chloé Villani, Conceptualization, Data curation, Formal analysis; Nir Hacohen, Conceptualization, Supervision, Project administration; Nathalie Abudi, Data curation, Writing - original draft; Rinat Abramovich, Conceptualization, Supervision, Methodology; Tamar Peretz, Supervision, Writing - review and editing; Andre Veillette, Conceptualization, Supervision, Writing - original draft, Writing - review and editing; Michal Lotem, Conceptualization, Supervision, Methodology, Writing - original draft, Project administration, Writing - review and editing

## Author ORCIDs

Emma Hajaj (iD) https://orcid.org/0000-0003-2437-3146
Thomas Eisenhaure (iD) http://orcid.org/0000-0003-3999-3540

## Ethics

Human subjects: Human samples were collected according to the approved IRB: Partners 2006-P-002051 in the Broad Institute of MIT and Harvard, Cambridge, Massachusetts.
Animal experimentation: Animal studies were approved by the Institutional Review Board - Authority for biological and biomedical models, Hebrew University, Jerusalem, Israel (MD-14602-5 and MD-15421-5).

## Decision letter and Author response

Decision letter https://doi.org/10.7554/eLife.52539.sa1
Author response https://doi.org/10.7554/eLife.52539.sa2

# Additional files

## Supplementary files

• Transparent reporting form

## Data availability

Data have been deposited to dbGaP under the accession code phs000815.v2.p1. To access these data users may apply for access to the dbGaP data repository (https://www.ncbi.nlm.nih.gov/books/NBK482114/).

The following dataset was generated:

| Author(s) | Year | Dataset title | Dataset URL | Database and Identifier |
| --- | --- | --- | --- | --- |
| Thomas E, Sarah | 2020 | Gene Expression and Regulatory | https://www.ncbi.nlm. | dbGaP, phs000815.v2. |

| EH, Alexandra CV, Nir H | Networks in Human Leukocytes - Immunological Variation Consortium | nih.gov/projects/gap/cgi-bin/study.cgi?study_id=phs000815.v2.p1 | p1 |

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
