## [Decision Letter]

Thank you for submitting your article "SLAMF6 deficiency augments tumor killing and skews towards an effector phenotype revealing it as a new T cell checkpoint" for consideration by *eLife*. Your article has been reviewed by Jeffrey Settleman as the Senior Editor, a Guest Reviewing Editor, and two reviewers. The reviewers have opted to remain anonymous.

The reviewers have discussed the reviews with one another and the Reviewing Editor has drafted this decision to help you prepare a revised submission.

Hajaj et al. reported that SLAMF6 plays an inhibitory role in the activation of tumor-specific CTL, suggesting SLAMF6 as an immune inhibitory receptor by using SLAMF6-KO/gp100 melanoma antigen-specific TCR transgenic T cells. The authors showed that SLAMF6 expression on human T cells was increased following T cell activation, although it is constitutively expressed on T cells. The SLAMF6 positive murine B16 melanoma specific T cells (Pmel-1) were significantly inhibited by SLAMF6 overly expressed melanoma cells than the mock melanoma cells, suggesting trans-acting effects. Pmel-1xSLAMF6 KO T cells that the authors newly generated had stronger functionality with more IFN-g and granzyme producing ability in vitro, and adoptive transfer of SLAMF6 KO T cells had stronger anti-tumor effects against the parental B16 melanoma. In terms of molecular mechanisms of action, decreased SAP adaptor protein in activated T cells with t-bet expression were found to be related to the increased T cell activity. In addition, persistent expression of LAG3 on the activated Pmel-1xSLAMF6 KO T cells was found, and addition of anti-LAG3 in vitro further increased IFN-g production of these T cells, suggesting of possible combination therapy with anti-LAG3 Ab and SLAMF6 KO T cells for cancer immunotherapy. The authors concluded that SLAMF6 is an inhibitory receptor and a possible target for cancer immunotherapy.

Essential revisions:

It was reported that SLAM6 reduces TCR signal strength in iNKT cells (Lu et al., 2019) and soluble SLAMF6 induces strong CD8^+^ T-cell effector function and improves anti-melanoma activity in vivo (Eisenberg et al., 2018). These previous studies demonstrated that SLAMF6 is a negative regulator of TCR signaling, and blockage of SLAMF6 signaling improves anti-tumor immunity. Therefore, basic negative roles on T cells of SLAMF6 and its potential interventions for cancer immunotherapy has previously been reported. However, detailed analyses of SLAMF6-deficient T cells specific for tumor antigen in this study, including trans-acting effect, involvement of SAP and T-bet, and specific relation to LAG3 expression, is interesting. Clarification of the current results and additional experiments for further mechanisms and possible clinical application may strengthen the impact of this study for possible publication.

1) It is still unclear whether inhibitory homotypic interaction of SLAMF6 for tumor-specific CTL is dependent on either cis or trans interaction, or both. In vitro and in vivo experiments to evaluate SLAMF6-WT or KO melanoma specific T cells against SLAMF6 overexpressed and mock B16-F10 melanoma should be performed. Experiments using SLAMF6-WT or KO APC incubated with SLAMF6-WT or KO gp100-specific TCR transgenic T cells should be performed.

2) Authors should examine T cell exhaustion status in the SLAMF6 knockout mice by using RNA-seq analysis. Since some studies suggested that SLAMF-family are expressed in pre-exhausted TIL cells which are considered as one of the targets for PD-1 Ab therapy (e.g., Nat Immunol. 201910:1335; Cancer Immunol Res 9:1485, 2019; Immunity 50:181, 2019).

3) For therapeutic point of view, authors should show anti-tumor effects of SLAMF6 knockout human T cells +/- anti-LAG3 Ab in vitro and in vivo xenogenic mouse study, and also evaluate anti-SLAM6 Ab +/- anti-LAG3 Ab in syngeneic mouse models. Confirmation of in vivo combination effects of SLAMF6 blockade and anti-LAG3 Ab is particularly important for this study.

4) In Figure 2E showing tumor growth data, and Figure 4H, I, J showing tumor growth and mouse survival data, it is not clear that the differences are significant or not. Appropriate statistical analyses should be performed.

5) In Figure 2, to make sure that SLAMF6 is responsible for the different anti-tumor effects, evaluation of additional SLAMF6 overexpressed tumor models, SLAM6 knock down, or neutralization of this effects by anti-SLAMF6 Ab, may be performed.

6) In Figure 3B, CD8^+^ T cell percentage in spleen of SLAMF6-KO x Pmel-1 mouse is lower than that of SLAMF6-WT x Pmel-1 mouse, suggesting that CD8^+^ T cell development may be impaired in SLAMF6-KO x Pmel-1 mouse thymus. Since SLAMF6 is reported to regulate NKT cell development, role of SLAMF6 on CTL development would be intriguing. Flow cytometric analysis of SLAMF6-KO x Pmel-1 mouse thymus may find new function of SLAMF6.

---

## [Author Response]

Essential revisions:It was reported that SLAM6 reduces TCR signal strength in iNKT cells (Lu et al., 2019) and soluble SLAMF6 induces strong CD8^+^ T-cell effector function and improves anti-melanoma activity in vivo (Eisenberg et al., 2018). These previous studies demonstrated that SLAMF6 is a negative regulator of TCR signaling, and blockage of SLAMF6 signaling improves anti-tumor immunity. Therefore, basic negative roles on T cells of SLAMF6 and its potential interventions for cancer immunotherapy has previously been reported. However, detailed analyses of SLAMF6-deficient T cells specific for tumor antigen in this study, including trans-acting effect, involvement of SAP and T-bet, and specific relation to LAG3 expression, is interesting. Clarification of the current results and additional experiments for further mechanisms and possible clinical application may strengthen the impact of this study for possible publication.

The references quoted by the reviewers certainly hint towards an inhibitory component in the SLAMF6 action, but do not demonstrate that this is the main and dominant aspect of the receptor in CD8 T cells. Our paper (Eisenberg et al., 2018) relates to T cell activation by ligand mimicry that can be interpreted as an independent agonist mechanism rather than a disclosure of the real nature of the SLAMF6 receptor. Lu et al. (2019) relates to iNKT development in primal stages. It was never shown that SLAMF6 is as strictly inhibitory in T cells as reflected in the SLAMF6^-/-^ mice experiments.

1) It is still unclear whether inhibitory homotypic interaction of SLAMF6 for tumor-specific CTL is dependent on either cis or trans interaction, or both. In vitro and in vivo experiments to evaluate SLAMF6-WT or KO melanoma specific T cells against SLAMF6 overexpressed and mock B16-F10 melanoma should be performed. Experiments using SLAMF6-WT or KO APC incubated with SLAMF6-WT or KO gp100-specific TCR transgenic T cells should be performed.

We thank the reviewers for suggesting completing this aspect of the research. Indeed, the data emerging from the added experiments now strengthen the inhibitory effect of SLAMF6 in a bilateral set of interactions.

Figures 5A-C were added to the revised manuscript and are described in subsection “The contribution of *cis* and *trans* SLAMF6 interactions to CD8^+^ T cell function”, and in the legend of Figure 5. The results show that the lack of SLAMF6 on the T cells or the target cells leads to a better response, suggesting that SLAMF6 plays a regulatory role both when expressed on the effectors (cis) and the targets (trans). This aspect is also dealt with in the Discussion section.

2) Authors should examine T cell exhaustion status in the SLAMF6 knockout mice by using RNA-seq analysis. Since some studies suggested that SLAMF-family are expressed in pre-exhausted TIL cells which are considered as one of the targets for PD-1 Ab therapy (e.g., Nat Immunol. 201910:1335; Cancer Immunol Res 9:1485, 2019; Immunity 50:181, 2019).

The murine model we used does not induce T cell exhaustion because it is an adoptive T cell transfer system, meant to evaluate the tumor destruction capacity of lymphocytes. Exhaustion is better evaluated in the context of chronic infection or chronic response to ongoing cancer, which is not the case here. However, checkpoint molecules that are linked to the exhaustion state were examined (Figure 6E), and among them, the most striking one was LAG-3. In the references mentioned by reviewers, SLAMF6 is considered an important player in pre-exhaustion, but this, per se, does not imply that SLAMF6 plays a negative or an agonist role.

3) For therapeutic point of view, authors should show anti-tumor effects of SLAMF6 knockout human T cells +/- anti-LAG3 Ab in vitro and in vivo xenogenic mouse study, and also evaluate anti-SLAM6 Ab +/- anti-LAG3 Ab in syngeneic mouse models. Confirmation of in vivo combination effects of SLAMF6 blockade and anti-LAG3 Ab is particularly important for this study.

Here, again, the reviewers’ comment contributed to the strength of the data. We performed an adoptive transfer experiment comparing SLAMF6 knockout T cells with or without the addition of an anti-murine LAG-3 Ab. Figure 6G-I show that the addition of LAG-3 blockade to ACT expedites tumor shrinkage compared to SLAMF KO T cells alone. Note that ACT with SLAMF6 KO T cells only is already very effective, but even so – it could be improved.

The revised data is in subsection “Mechanism associated with the inhibitory function of SLAMF6” and in the legend of Figure 6, and is dealt with in the Discussion section. Regarding the effect of anti-SLAM6 Abs with or without the addition of anti-LAG-3 Abs, Yigit et al. (2018) have demonstrated an agonistic effect of anti-SLAMF6 (NT7, Biolegend). However, in our experimental systems, this antibody did not improve the T cell response, and therefore we did not use it.

4) In Figure 2E showing tumor growth data, and Figure 4H-J showing tumor growth and mouse survival data, it is not clear that the differences are significant or not. Appropriate statistical analyses should be performed.

We added this information to the figures and the text.

5) In Figure 2, to make sure that SLAMF6 is responsible for the different anti-tumor effects, evaluation of additional SLAMF6 overexpressed tumor models, SLAM6 knock down, or neutralization of this effects by anti-SLAMF6 Ab, may be performed.

The Pmel-1 adoptive transfer model is considered the best experimental model to capture the natural situation of the CD8 T cell response against a native tumor-derived antigen. For this reason, we adhered to this model. However, in addition, we evaluated the effect of SLAMF6 in EL4 cells, a thymoma cell line derived from a C57BL/6 (H-2b) mouse, ectopically expressing SLAMF6.

6) In Figure 3B, CD8^+^ T cell percentage in spleen of SLAMF6-KO x Pmel-1 mouse is lower than that of SLAMF6-WT x Pmel-1 mouse, suggesting that CD8^+^ T cell development may be impaired in SLAMF6-KO x Pmel-1 mouse thymus. Since SLAMF6 is reported to regulate NKT cell development, role of SLAMF6 on CTL development would be intriguing. Flow cytometric analysis of SLAMF6-KO x Pmel-1 mouse thymus may find new function of SLAMF6.

We agree that T cell development in the absence of SLAMF6 is interesting; however, it was beyond the scope of this work.